# Learn, Imagine and Create: Text-to-Image Generation from Prior Knowledge

**Tingting Qiao**[1,2*]   **Jing Zhang**[2*]   **Duanqing Xu**[1†]   **Dacheng Tao**[2]

[1]College of Computer Science and Technology, Zhejiang University, China
[2]UBTECH Sydney AI Centre, School of Computer Science, Faculty of Engineering
The University of Sydney, Darlington, NSW 2008, Australia
`{qiaott,xdq}@zju.edu.cn, {jing.zhang1,dacheng.tao}@sydney.edu.au`

## Abstract

Text-to-image generation, *i.e.* generating an image given a text description, is a very challenging task due to the significant semantic gap between the two domains. Humans, however, tackle this problem intelligently. We learn from diverse objects to form a solid prior about semantics, textures, colors, shapes, and layouts. Given a text description, we immediately imagine an overall visual impression using this prior and, based on this, we draw a picture by progressively adding more and more details. In this paper, and inspired by this process, we propose a novel text-to-image method called *LeicaGAN* to combine the above three phases in a unified framework. First, we formulate the *multiple priors learning phase* as a textual-visual co-embedding (TVE) comprising a text-image encoder for learning semantic, texture, and color priors and a text-mask encoder for learning shape and layout priors. Then, we formulate the *imagination phase* as multiple priors aggregation (MPA) by combining these complementary priors and adding noise for diversity. Lastly, we formulate the *creation phase* by using a cascaded attentive generator (CAG) to progressively draw a picture from coarse to fine. We leverage adversarial learning for LeicaGAN to enforce semantic consistency and visual realism. Thorough experiments on two public benchmark datasets demonstrate LeicaGAN's superiority over the baseline method. Code has been made available at `https://github.com/qiaott/LeicaGAN`.

## 1   Introduction

Text-to-image (T2I) generation aims to generate a semantically consistent and visually realistic image conditioned on a textual description. This task has recently gained a lot of attention in the deep learning community due to both its significant relevance in a number of applications (such as photo editing, art generation, and computer-aided design) and its challenging nature, mainly due to the semantic gap between the domains and the high dimensionality of the structured output space.

Prior methods addressed this problem by first using a pre-trained text encoder to obtain a text feature representation conveying the relevant visual information of a given text description. Then, this text feature representation was served as input to generative neural networks (GANs) [5] to create an image which visually matches the semantic content of the input text [23, 44, 37, 20]. Reed *et al.* proposed using a deep convolutional and a recurrent text encoder together with generative networks [23] for this purpose. In [44], the same text encoder was used and several GANs were stacked to progressively generate more detailed images. Similar text encoders were also utilized in [37, 20],

---

[*]indicates equal contribution.

[†]corresponding author

with Xu *et al.* adding an attention mechanism to condition different sub-regions of the image on words which are relevant to those regions [37], while Qiao *et al.* proposed a mirror structure by leveraging an extra caption model to enforce semantic consistency between the generated image and the given text description [20].

Although impressive results have been obtained using these methods, they share a common limitation, namely that the generator relies on a single text encoder to extract the embedded visual information. On the one hand, the visual space is high dimensional and structured, so it is hard to extract a visual vector covering many different aspects like low-level textures and colors and high-level semantics, shapes, and layouts. On the other hand, an image is much more informative than a piece of text, indeed, '*a picture is worth a thousand words*'. Therefore, it is challenging to embed text and image into a common semantic space. We hypothesize that this limitation could be overcome by introducing several semantic subspaces in which we decompose the image and respectively co-embedding the decompositions with the text.

Going one step further, we analyze how humans achieve this goal. As humans, when we are asked to draw a picture given a text description (for instance, '*a small bird with blue wings and with a white breast and collar*'), we first build a coarse mental image about the core concept of '*a bird*' before enriching this initial mental image by progressively adding more details based on the given text; in this case, for instance, the color of the wings and the breast. It is noteworthy that building this mental image about the core concept is not a trivial process since it requires us to have learned a rich prior about literal concepts, semantics, textures, colors, shapes and layouts of diverse objects. Taking the online drawing game *Quick Draw* [21] developed by Jongejan *et al.* as an example, when people from different countries draw a picture given a concept word, although there are some differences between these drawings, they all share a common underlying appearance, *i.e.* the aforementioned coarse mental image [22]. Additionally, the studies in [2, 19] identified two critical concepts termed *visual realism* and *intelligence realism*, wherein the latter explaining the phenomenon by which a child's drawing may not be visually realistic because children just draw something based on what they know, thereby conveying the core concept about an object.

Inspired by these studies, here we propose a novel T2I method called *LeicaGAN* to combine the above "LEarn, Imagine and CreAte" phases in a unified adversarial learning framework. First, we formulate the *multiple priors learning phase* as textual-visual co-embedding (TVE) comprising a text-image encoder for learning semantics, textures and colors priors, and a text-mask encoder for learning shape and layout priors. Then, we formulate the *imagination phase* as multiple priors aggregation (MPA) by combining the previous complementary priors together and adding noise for diversity. Lastly, we formulate the *creation phase* by using a cascaded attentive generator (CAG) to progressively draw a picture in a coarse to fine manner. We leverage adversarial learning for LeicaGAN to enforce semantic consistency and visual realism. The proposed method is evaluated on two public benchmark datasets, namely CUB and Oxford-102. Both quantitative and qualitative results demonstrate the superiority of LeicaGAN over the representative baseline method.

The main contributions of this work are as follows. First, we tackle the T2I problem by decomposing it into three phases: multiple priors learning, imagination and creation - thereby mimicking how humans solve this task. Second, we propose a novel method named LeicaGAN which includes a textual-visual co-embedding network (TVE), a multiple priors aggregation network (MPA) and a cascaded attentive generator (CAG) to respectively formulate the aforementioned three phases in a unified framework trained via adversarial learning. Third, thorough experiments on two public benchmark datasets demonstrate the effectiveness of the employment of the idea of LeicaGAN.

## 2   Related work

**Text-to-Image generation**. Generative adversarial networks (GANs) [6] have been extensively used for image generation conditioned on discrete labels [15, 17], images [10, 47, 39] and text [23, 44, 37, 46]. Reed *et al.* first proposed conditional GANs for T2I generation [23]. This work was extended by stacking several attention-based GANs and generating images in multi-steps [44, 45, 37]. Zhang *et al.* adopted a hierarchically-nested framework in which multi-discriminators were used for different layers of the generator [46]. These works have in common that a single text encoder was used to obtain text embeddings. Another popular approach has been to provide more information for image generation [9, 7, 11]. For example, Hong *et al.* added a layout generator that predicted the

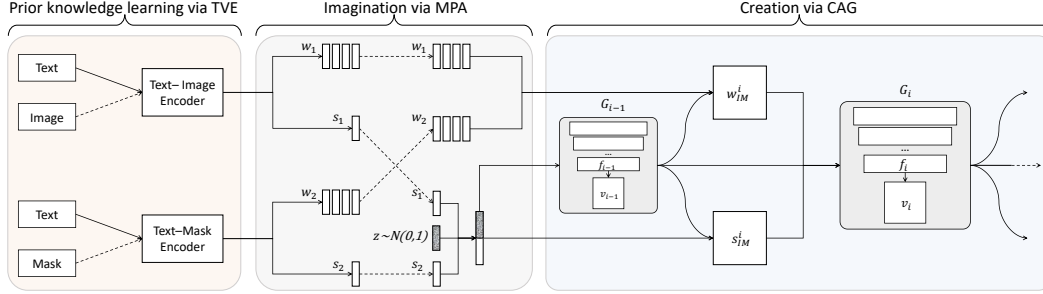

Figure 1: The LeicaGAN framework, which tackles the T2I problem by decomposing it into three phases, which are 1. multiple priors learning via text-visual co-embedding (TVE), 2. imagination via multiple priors aggregation (MPA) and 3. creation via a cascaded attentive generator (CAG).

bounding boxes and shapes of objects [9] and a similar idea was adopted in [7]. Johnson *et al.* built up a scene graph dataset that aimed to provide clear layout information for the target image [11]. In contrast to these methods, we focus on generating an image only conditioned on a text description, from which we extract and aggregate different visual priors based on multiple text-encoders.

**Attention generative model**. Attention mechanisms, as one of the most influential ideas in the deep learning community, have become an integral part of generative models because of the fact that they can be conveniently modelled, *e.g.* spatially in images, temporally in language or even in multi-modal generation. They also boost deep model performance by guiding the generators to focus on the relevant information [43, 42, 40, 25, 37, 13, 3, 26, 12, 14, 32]. In this spirit, we also adopt an attention mechanism in LeicaGAN to help the generators decide which parts of the textual information to focus on when respectively refining the relatively coarse image from the previous step.

**Multi-modal learning**. The proposed textual-visual co-embedding method falls into the category of pairwise multi-modal learning [4, 8, 30, 48]. In particular, our approach is motivated: $(i)$ by the learning process which focuses on individual pairs of samples and learning objectives, *e.g.* the variants of the correlation loss [4]; and $(ii)$ by the adversarial learning methods, especially with respect to using an adversarial loss to reduce the domain gap between the text and visual input [34, 41]. Specifically, we propose two textual-visual encoders to co-embed text-image and text-mask pairs into two common subspaces in the multiple priors learning phase, which map the text to visual semantics, textures, colors, shapes, and layouts accordingly.

## 3 LeicaGAN for Text-to-Image Generation

Given a text description $t = \{u_1, u_2 \ldots u_L\}$ consisting of $O$ words $u$, the goal of T2I generation is to learn a mapping function to convert $t$ to a corresponding visually realistic image $\widehat{v}$. We propose LeicaGAN to tackle this problem, which includes an initial multiple priors learning phase, an imagination phase, and a creation phase, which are shown in Figure 1 and presented in details below.

### 3.1 Multiple priors learning via Text-Visual co-Embedding (TVE)

Co-embedding textual-visual pairs in a common semantic space enables the text embeddings to convey the visual information needed for the following image generation. A textual-visual co-embedding model is trained with dataset $S = \{(t_n, v_n, c_n), n = 1, 2, ...N\}$, where $t \in T$ represents a text description, $v \in V$ represents visual information, which may be an image $v^I$ or a segmentation mask $v^S$, and $c \in C$ represents a class label. The TVE model consists of a text encoder $E_T$ and an image encoder $E_V$. $E_T$ employs a recurrent neural network (RNN) [28] to encode the input text into a word-level textual feature $w$ and a sentence-level textual feature $s$. $E_V$ employs a convolutional neural network (CNN) [31] to encode the visual information into a local visual feature $l$ and a global visual feature $g$. Mathematically,

$$w, s = E_T(t); \quad l, g = E_V(v), \tag{1}$$

where $w \in \mathbb{R}^{D \times O}$ is the concatenation of the $O$ hidden states of RNN while $s \in \mathbb{R}^D$ is the last hidden state, $l \in \mathbb{R}^{D \times H}$ is extracted from an intermediate layer of the CNN while $g \in \mathbb{R}^D$ is obtained from the last pooling layer, $D$ is the dimension of the embedding space, and $H$ is the feature map size.

**Text-Image Encoder** ($E_T^I$). To project the input text $t$ and the image $v^I$ to the same common semantic space, we leverage an attentive model to calculate the similarity between a textual feature ($w$ or $s$) and a visual feature ($l$ or $g$). The similarity matrix $s_{w|l}$ for all possible pairs of words in the sentence and sub-regions in the image is calculated by

$$\overline{s}_{w|l} = softmax_H(l^T w), \tag{2}$$

where $\overline{s}_{w|l} \in \mathbb{R}^{H \times O}$ and $softmax_H(\cdot)$ indicates a normalization operation via a softmax function calculated along the $H$-dimension. Then the word-level feature and local visual feature are fed into an attention module, in which the weighted visual feature is calculated as:

$$\hat{l} = l \cdot softmax_O(\alpha_1 \overline{s}_{w|l}) \tag{3}$$

where $\alpha_1$ is a smoothing factor and $softmax_O$ indicates a softmax function calculated along the $O$-dimension. Then the local-level image-text matching score between $t$ and $v^I$ is obtained:

$$s_{w|l} = log(\sum_{o=1}^{O} exp(\alpha_2 \, cos(\hat{l}, w)))^{\frac{1}{\alpha_2}}, \tag{4}$$

where $\alpha_2$ is a smoothing factor and $cos(\cdot)$ represents the cosine similarity between the vectorization of $\hat{l}$ and $w$ along the $D$-dimension. For a batch of text-image pairs, the posterior probability of $t$ matching with $v^I$ is defined as:

$$p(w|l) = exp(\alpha_3 s_{w|l}) \Big/ \sum_B exp(\alpha_3 s_{w'|l'}), \tag{5}$$

where $\alpha_3$ is a smoothing factor and $B$ is the batch size. Then, we can minimize the negative log posterior probability that the images are matched with their corresponding text descriptions as follows:

$$\mathcal{L}_{w|l} = -\frac{1}{N} \sum_{n=1}^{N} log \, p(w_n|l_n). \tag{6}$$

Symmetrically, we also minimize the $\mathcal{L}_{l|w} = -\frac{1}{N} \sum_{n=1}^{N} log \, p(l_n|w_n)$ to match text descriptions with images. Moreover, we also calculate the similarity between sentence-level text and global image feature pairs $(s, g)$ and minimize $\mathcal{L}_{s|g}$ and $\mathcal{L}_{g|s}$ likewise. The final similarity loss $\mathcal{L}_{sim}$ is defined as:

$$\mathcal{L}_{sim} = \mathcal{L}_{w|l} + \mathcal{L}_{l|w} + \mathcal{L}_{s|g} + \mathcal{L}_{g|s}. \tag{7}$$

Following the common practise [35, 38], we then employ a triplet loss to make the images belonging to the same category can be embedded closely. Specially, we use global visual features to calculate the triplet loss:

$$\mathcal{L}_{triplet} = -\frac{1}{N} \sum_{n=1}^{N} max(\| g - g_p \|^2 - \| g - g_n \|^2 + \beta_1, 0) \tag{8}$$

where $max(\cdot, 0)$ is the hinge loss function, $g_p$ and $g_n$ are the global features of the randomly sampled positive and negative samples, $\beta_1$ represents the violate margin.

Additionally, since images and text belong to different domains, it is difficult to directly project them into the same feature space [18, 34, 41]. To reduce this domain gap, we adopt domain adversarial learning proposed in [34] to adapt each domain to an underlying common domain. A modality classifier $D_{modal}$ is applied to detect the real modality of the input, while the encoders try to fool $D_{modal}$ by projecting the input into the underlying domain where paired text and image are indistinguishable. The domain adversarial loss $\mathcal{L}_{adv}$ is defined as:

$$\mathcal{L}_{adv} = -\frac{1}{N} \sum_{n=1}^{N} \mathcal{L}_{GT} \cdot (log \, D_{modal}(g_n) + log \, D_{modal}(1 - s_n)), \tag{9}$$

where $\mathcal{L}_{GT}$ is a one-hot vector indicating the ground-truth modality label and $D_{modal}(\cdot)$ is the predicted modality probability of each input. The final loss $\mathcal{L}_{TI}$ for the $E_T^I$ is defined as:

$$\mathcal{L}_{TI} = \gamma_1 \mathcal{L}_{sim} + \gamma_2 \mathcal{L}_{triplet} + \gamma_3 \mathcal{L}_{adv}, \tag{10}$$

where $\gamma_1$, $\gamma_2$ and $\gamma_3$ are the loss weights.

**Text-Mask Encoder** ($E_T^M$). To strengthen text embeddings conveying more shape and layout prior, we also construct a text-mask encoder like the text-image encoder. They differ in the visual input

where a segmentation mask $v^S$ is used instead of an image $v^I$. Likewise, we train the text-mask encoder by minimizing the following loss function:

$$\mathcal{L}_{TM} = \gamma_4 \mathcal{L}_{sim}^{TM} + \gamma_5 \mathcal{L}_{cls} + \gamma_6 \mathcal{L}_{adv}^{TM}, \tag{11}$$

where $\mathcal{L}_{sim}^{TM}$ and $\mathcal{L}_{adv}$ are the same loss functions defined in Eq. (7) and Eq. (9), $\gamma_4$, $\gamma_5$ and $\gamma_6$ are the loss weights. The classification loss $\mathcal{L}_{cls}$ is defined as:

$$\mathcal{L}_{cls} = -\frac{1}{N} \sum_{n=1}^{N} log\ p(c_n|s_n) + log\ p(c_n|g_n). \tag{12}$$

## 3.2 Imagination via Multiple Priors Aggregation (MPA)

In the multiple priors learning phase, we obtain two types of text embeddings from the text-image encoder $E_T^I$ and text-mask encoder $E_T^M$ respectively conveying visual information about the semantics, textures and colors, and shapes and layouts. To mimic the humans' imagination process, we aggregate the learned priors within the two encoders given a text description $t$. It is formulated as

$$w_I, s_I = E_T^I(t); \quad w_M, s_M = E_T^M(t), \tag{13}$$

where $w_i \in \mathbb{R}^{D \times O}$ and $s_i \in \mathbb{R}^D$, $i \in \{I, M\}$. Then, we fuse the sentence-level embeddings as $s_{IM} = [W_I^s s_I, W_M^s s_M]$, where $[\cdot]$ denotes the concatenate operation and $W_I^s, W_M^s \in \mathbb{R}^{\frac{K}{2} \times D}$ are transformation matrices. After the fusion process, we obtain the mental image as: $\{z, s_{IM}, w_I, w_M\}$. $z \in \mathbb{R}^K$ is a random noise sampled from a Gaussian distribution for diversity.

## 3.3 Creation via Cascaded Attentive Generators (CAG)

After obtaining the mental image in the imagination phase, we begin to draw it out in the creation phase. However, combining all the relevant information to generate a photo-realistic image with correct semantics is challenging. Carefully designed network architectures are critical to achieve a good performance [44, 37, 46, 36, 43]. In this paper, we use the cascaded attentive generative network [44, 37] to address this challenge.

**Initial coarse image generation**. In the first step, we feed the input $U_0 = [z, s_{IM}]$ into a generator $G_0$ to obtain an initial coarse image $\widehat{v_0}$:

$$\widehat{v_0} = G_0\left(U_0\right). \tag{14}$$

**Attentive feature generation**. During drawing, we humans enrich the coarse sketch with more and more details by attending to specific regions. To mimic this process, we design an attention feature generation module which produces two attentive word- and sentence-context features $w_{IM}^i, s_{IM}^i$ by fusing the two pairs of textual features, i.e. $(w_I, w_M)$ and $(s_I, s_M)$, with the visual feature $f_{i-1}$ of the previously generated image $\hat{v}_{i-1}$. Mathematically, this is formulated as:

$$w_{IM}^i = \sum_{j \in \{I, M\}} \delta_j \left( w_j^i \left( softmax \left( w_j^{i^T} f_{i-1} \right) \right) \right), \tag{15}$$

where $w_j^i$ is the word embedding after a perception layer, i.e. $w_j^i = P_i^j w_j$, $P_i^j \in \mathbb{R}^{X_i \times D}$ and $j \in \{I, M\}$, $f_{i-1} \in \mathbb{R}^{X_i \times Y_i}$ is the feature map from an intermediate layer of the $G_{i-1}$. $\delta_I$ and $\delta_M$ are two weights subjected to $\delta_I + \delta_M = 1$. Then, an attentive sentence feature is also learned to provide a global guidance to the generators. Mathematically, this is formulated as:

$$s_{IM}^i = \hat{s}_{IM}^i \circ \left( softmax \left( f_{i-1} \circ \hat{s}_{IM}^i \right) \right), \tag{16}$$

where $\hat{s}_{IM}^i$ is the sentence embedding after a perception layer, i.e. $\hat{s}_{IM}^i = Q_i s_{IM}$, $Q_i \in \mathbb{R}^{X_i \times K}$, and $\circ$ denotes the element-wise multiplication.

**Image refinement via cascaded attentive generative networks**. After obtaining the attentive word- and sentence-context features, we input them with the image feature $f_{i-1}$ together to the $i^{th}$ generator $G_i$, i.e. $U_i = [f_{i-1}, s_{IM}^i, \lambda_w w_{IM}^i]$, where $\lambda_w$ is a weight factor, to produce the $i^{th}$ image:

$$\hat{v}_i = G_i\left(f_{i-1}, U_i\right), i \in \{1, 2, ...\}. \tag{17}$$

Images are progressively generated in these generators in a coarse-to-fine manner.

## 3.4 Objective function

We leverage adversarial training on each $G_i$ of LeicaGAN. As shown in the current state-of-the-art [37, 46, 20], a carefully designed adversarial loss function is essential for stable training and optimal performance. We therefore employ two adversarial losses: a visual realism adversarial loss to ensure that the generators generate visually realistic images, and a text-image pair-aware adversarial loss to guarantee the semantic consistency between the input text and the generated image, $i.e.$,

$$\mathcal{L}_{G_i} = -\frac{1}{2}\mathbb{E}_{\hat{v}_i \sim p_{\hat{v}_i}}\left[\log\left(D_i\left(\hat{v}_i\right)\right)\right] - \frac{1}{2}\mathbb{E}_{\hat{v}_i \sim p_{\hat{v}_i}}\left[\log\left(D_i\left(\hat{v}_i, t\right)\right)\right]. \tag{18}$$

We further use $\mathcal{L}_{sim}$ defined in Eq. (7) to constrain the generated images $\hat{v}_i$ to share the same semantics as the input text description $t$. It is noteworthy that the network weights of $E_T$ are kept fixed while training the generators. The final objective function of the generator $G$ is defined as:

$$\mathcal{L}_G = \sum_{i=0}^{m-1}\mathcal{L}_{G_i} + \mathcal{L}_{sim}^i(\hat{v}_i, t), \tag{19}$$

where $m$ is the number of generators. Accordingly, the discriminator $D_i$ is trained by minimizing the following loss:

$$\begin{aligned}\mathcal{L}_{D_i} = &-\tfrac{1}{2}\mathbb{E}_{v_i \sim p_{v_i}}\left[\log\left(D_i\left(v_i\right)\right)\right] - \tfrac{1}{2}\mathbb{E}_{\hat{v}_i \sim p_{\hat{v}_i}}\left[\log\left(1 - D_i\left(\hat{v}_i\right)\right)\right] \\ &-\tfrac{1}{2}\mathbb{E}_{v_i \sim p_{v_i}}\left[\log\left(D_i\left(v_i, t\right)\right)\right] - \tfrac{1}{2}\mathbb{E}_{\hat{v}_i \sim p_{\hat{v}_i}}\left[\log\left(1 - D_i\left(\hat{v}_i, t\right)\right)\right]\end{aligned} \tag{20}$$

The final objective function of the discriminator $D$ is defined as:

$$\mathcal{L}_D = \sum_{i=0}^{m-1}\mathcal{L}_{D_i}. \tag{21}$$

# 4 Experiments

## 4.1 Experiment settings

**Datasets**. We evaluated our model on two commonly used datasets, $i.e.$ the CUB bird [33] and Oxford-102 flower [16]. In contrast to previous works [44, 37], which processed these datasets into class-disjoint training and testing sets, we randomly re-split them to ensure both training and testing sets contain images from all classes resulting in two class-balanced datasets: CUB$^*$ containing 8,855 training and 2,933 testing data belonging to 200 categories, and Oxford$^*$ containing 7,034 training and 1,155 testing data belonging to 102 categories. Each image in both datasets has 10 text captions.

**Evaluation metrics**. Following the common practice [44, 37, 20], the Inception Score [27] was used to measure both the objectiveness and diversity of the generated images. The inception models provided by [29] were used for testing models trained on the CUB$^*$ and Oxford-102$^*$. Additionally, the R-precision introduced in [37] was used to evaluate the visual-semantic similarity between the generated images and their corresponding text descriptions, We reported the precision score of top-5.

**Implementation details**. Following [37, 20], the text encoder $E_T$ was a pre-trained bi-directional LSTM [28] and the image encoder $E_V$ was built upon the Inception-v3 model [31]. The visual local features were obtained from the $mixed\_6e$ layer. The dimension $D$ was 256, the sentence length $O$ was 18 and the image region size $H$ was $299 \times 299$. The generator consisted of the proposed attention module, two residual blocks, and an upsampling module followed by a convolutional layer. The discriminators adopted the structure in [44]. The visual embedding dimension $X_i$ was set to 32, $Y_i = q_i^2$, where $q_i$ was 64, 128, and 256 for the three stages. $\alpha_1, \alpha_2, \alpha_3$ and $\lambda_w$ were set to 4, 5, 10, 1. The balance weights $\delta^I = 0.8$ and $\delta^M = 0.2$. The weights for training TVE of LeicaGAN with the best performance on the CUB bird dataset were $\gamma_1 = 1, \gamma_2 = 1, \gamma_3 = 4, \gamma_4 = 1, \gamma_5 = 1, \gamma_6 = 0.5$. On the Oxford-102 flower dataset, the best weights were $\gamma_1 = 1, \gamma_2 = 1, \gamma_3 = 0, \gamma_4 = 1, \gamma_5 = 0.5, \gamma_6 = 0$. Please see the appendix for more implementation details.

## 4.2 Main Results

**Objective comparisons**. To intuitively verfiy the effectiveness of the idea of LeicaGAN of employing different text encoders, we chose the state-of-the-art T2I methods AttnGAN [37] as our baseline model, as it only employs one text encoder and shares the similar structure of the generators. Table

Table 1: Inception Score results comparsion between AttnGAN and LeicaGAN on the original splits and new splits of CUB and Oxford-102 datasets.

| Model | CUB | Oxford-102 | CUB* | Oxford-102* |
|---|---|---|---|---|
| GAN-INT-CLS [23] | 2.88±0.04 | 2.66±0.03 | - | - |
| GAWWN [24] | 3.62±0.07 | - | - | - |
| StackGAN [44] | 3.70±0.04 | 3.20±0.01 | - | - |
| StackGAN++ [45] | 4.04±0.05 | - | - | - |
| AttnGAN [37] | 4.36±0.03 | 3.75±0.02 | 5.45±0.06 | 3.57±0.02 |
| LeicaGAN | **4.62±0.06** | **3.92±0.02** | **5.69±0.06** | **3.80±0.01** |

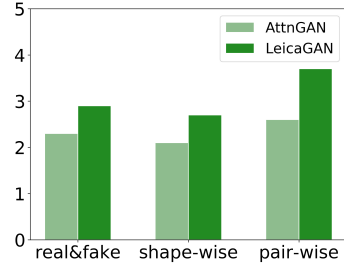

Figure 2: Human study results.

Table 2: Inception Score and R-precision results of LeicaGAN with different weight settings.

| Evaluation Metric | Inception Score | | R-precision | |
|---|---|---|---|---|
| | CUB* | Oxford-102* | CUB* | Oxford-102* |
| AttnGAN (Baseline) | 5.45±0.06 | 3.57±0.01 | 81.45 | 82.33 |
| LeicaGAN, w/o $E_T^M$ | 5.60±0.05 | 3.68±0.01 | 82.95 | 85.03 |
| LeicaGAN, $\lambda_w$=0 | 5.63±0.04 | 3.73±0.02 | 84.10 | 85.77 |
| LeicaGAN, $\delta_I$=0.2 | 5.36±0.04 | 3.59±0.02 | 81.21 | 82.09 |
| LeicaGAN, $\delta_I$=0.4 | 5.39±0.05 | 3.50±0.01 | 81.37 | 82.53 |
| LeicaGAN, $\delta_I$=0.6 | 5.47±0.04 | 3.65±0.01 | 82.84 | 84.72 |
| LeicaGAN, $\delta_I$=0.8 | **5.69±0.06** | **3.80±0.01** | **85.28** | **85.81** |
| LeicaGAN, $\delta_I$=1.0 | 5.55±0.06 | 3.75±0.02 | 81.11 | 83.89 |

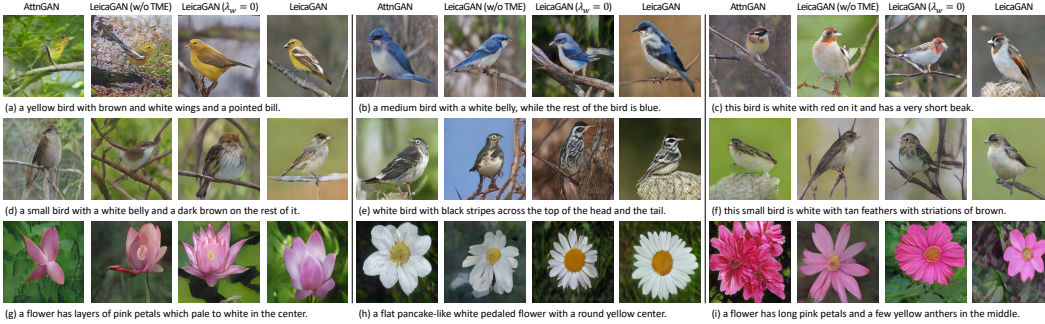

(a) a yellow bird with brown and white wings and a pointed bill. (b) a medium bird with a white belly, while the rest of the bird is blue. (c) this bird is white with red on it and has a very short beak.

(d) a small bird with a white belly and a dark brown on the rest of it. (e) white bird with black stripes across the top of the head and the tail. (f) this small bird is white with tan feathers with striations of brown.

(g) a flower has layers of pink petals which pale to white in the center. (h) a flat pancake-like white pedaled flower with a round yellow center. (i) a flower has long pink petals and a few yellow anthers in the middle.

Figure 3: Examples of images generated by AttnGAN and LeicaGAN with different weight settings.

1 shows the performance of both models on both datasets. As it can be seen, LeicaGAN achieved the higher Inception Score and outperformed the baseline model by large margins, indicating the effectiveness of employing two text encoders and showing that LeicaGAN can generate more diverse images with better quality and semantic consistency.

In addition, to validate the effectiveness of various component choices of our method, we conducted several comparative experiments by excluding/including these components. The results are shown in Table 2. First, the weight $\delta_I$ in Eq. (15) used to balance the importance of the attentive features from the different text encoders was studied. LeicaGAN $\delta_I = 0.8$ achieved the best performance, showing that the word features from both $E_T^I$ and $E_T^M$ had a positive impact on generator performance. The text features from $E_T^I$ had a higher weight as they provided more essential visually related information to the generators, $e.g.$ color, texture and semantic information. Then, comparing LeicaGAN with and without $E_T^M$, we can see that the employment of $E_T^M$ helped LeicaGAN achieve a higher Inception Score and R-precision on both datasets, demonstrating that the text features from $E_T^M$ indeed provided extra visual information to the generators. Additionally, we also tested LeicaGAN with ($\lambda_w$=0) in which only global text embeddings from $E_T^I$ and $E_T^M$ were used for the image generation. As shown in Table 2, employing both local and global embeddings collaboratively led to significant performance gains on both datasets. These results indicate the effectiveness of the employment of the collaborative local and global attention model in the generators of LeicaGAN.

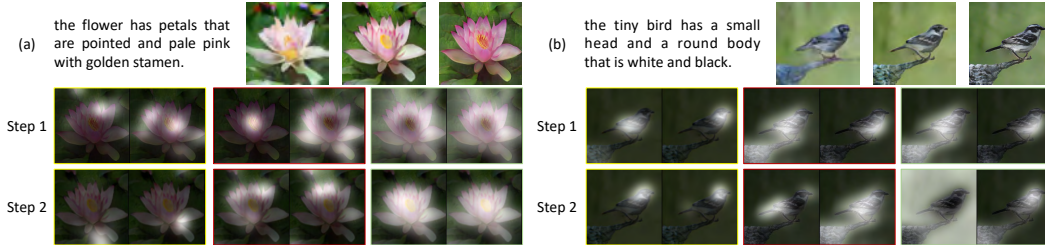

Figure 4: Visualization of attention maps and images generated by LeicaGAN. The first row shows the generated images in three steps. The attention maps shown in yellow, red and green frames correspond to intermediate attention for producing $w_I^i$, $w_M^i$ and $\hat{s}_{IM}^i$ in Eq. (15) and (16).

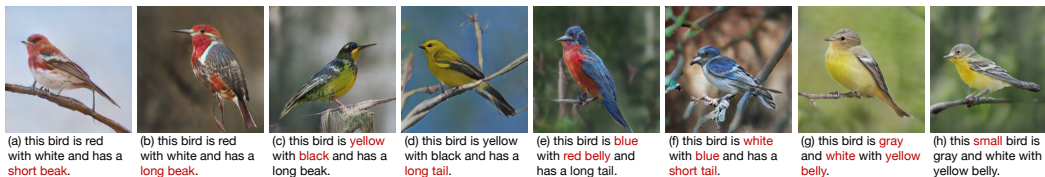

Figure 5: Images generated by LeicaGAN by modifying some words of the input text descriptions.

**Subjective visual comparisons**. Subjective visual comparisons between the AttnGAN and Leica-GAN with different weight settings are presented in Figure 3. It can be seen that some images from AttnGAN have relatively unsatisfactory shapes and details, *e.g.* images generated by AttnGAN in Figure 3 (a,c,f,g). Similarly, images generated by LeicaGAN without $E_T^M$ in Figure 3 (d) also lack some important details about shape and texture and have a strange shape and blurry textures. However, by using global features from $E_T^M$, LeicaGAN ($\lambda_w$=0) generated images with visually better shapes, colors and more details, *e.g.* the results generated by LeicaGAN ($\lambda_w$=0) in Figure 3 (b, c, g, h, i), demonstrating the superiority of utilizing the complementary $E_T^I$ and $E_T^M$ to encode semantic, color, texture and shape information simultaneously. Furthermore, compared with LeicaGAN ($\lambda_w$=0), LeicaGAN leveraged both local and global attentive features to produce semantically consistent images with more details.

**Visual inspection of the generated results**. To better understand the cascaded generation process of LeicaGAN, we visualized both the intermediate images and the attention maps corresponding to $w_I^i$, $w_M^i$, and $\hat{s}_{IM}^i$, $i = \{1, 2\}$, in Eq. (15), which are shown in Figure 4 (a, b). As it can be seen, similar to human practice, coarse images were generated in the first stage, which only had abstract shapes and colors. Gradually, with more visual information from $E_T^I$ and $E_T^M$, the subsequent generators could draw relatively fine-grained images by filling in more details and refining the shapes. The attention maps of $w_I^i$ obtained by fusing the visual feature and text feature provided by $E_T^I$ led the generator to focus on semantic details, such as the flower petals in Figure 4 (a) or the feathers and the wings of the bird in Figure 4 (b). By contrast, the attention maps of $w_M^i$ focused on the shape and layout of the objects, like the bodies of the flowers and birds. The attention maps of $s_{IM}^i$ obtained by fusing global features from $E_T^I$ and $E_T^M$ mainly focused on the global layout or background and served as strong guidance for the generators. Additionally, we also present the images generated by LeicaGAN with the modification of some words of the input text descriptions in Figure 5. As it can be seen, LeicaGAN is able to accurately capture the semantic differences in the text description.

**Human perceptual study**. We also performed human study to investigate the visual qualities of our generated results compared with AttnGAN [37]. For each method, we first randomly sampled 50 images from both the generated bird and flower samples. They were randomly shown one by one to the participants (up to 60) who were asked to score the images on a scale of 0 (worst) to 5 (best) with respect to the criteria of whether participants could regarded the generated images as realistic (*Real&Fake Test*), whether the generated images had a good and reasonable shape (*Shape-Wise Test*) and whether the generated images were semantically consistent with the input text (*Pair-Wise Test*). The results of these three tests are illustrated in Figure 2, which shows that our method outperformed AttnGAN in terms of authenticity, object shape in the images, and semantic consistency.

### 4.3 Limitations and discussion

Although LeicaGAN showed superiority in generating visually realistic and semantically consistent images, it still has some limitations. First, we only used $E_T^I$ and $E_T^M$ to learn semantics, textures, colors, shape and layout priors in the MPA phase. There is other prior that could be further taken into consideration, $e.g.$ fine-grained attributes. Second, it would be valuable to explore efficient and diverse modules to strengthen the impact of imagination. Third, the TVE models were trained separately from the MPA and CAG models in the current implementation. We believe they could be trained end-to-end to further enhance the performance.

In addition, although the Inception Score and R-precision are regarded as effective and are commonly-used for evaluating the diversity of the generated images and the semantic consistency between the input text and the generated images, while carrying out experiments, it was observed that a text-to-image model could get a fairly high performance even when the model actually generated visually-low quality images, see e.g. Figure 1 in [1]. Therefore, a more objective evaluation method is also worthy to be explored in the future work.

## 5 Conclusion

In this work, we introduce a novel T2I method called LeicaGAN to mimic how humans solve the T2I problem. We first decompose T2I into three sequential phases: a multiple priors learning phase, an imagination phase, and a creation phase. Accordingly, in the first phase, we propose a text-image encoder for learning semantic, texture, and color priors and a text-mask encoder for learning shape and layout priors. In the second phase, we combine these complementary priors from both encoders and add noise for diversity to mimic imagination. Lastly, we propose a cascaded attentive generator to leverage both local and global features, progressively generating the output image. Adversarial learning was utilized to train LeicaGAN and enforce semantic consistency and visual realism. Both objective and subjective experiment results demonstrate the effectiveness of the proposed LeicaGAN.

**Acknowledgments**: This work was supported in part by the National Natural Science Foundation of China Project 61806062, the key provincial R&D project of Zhejiang Province 2019C03137, the the Science and Technology Project of Cultural Relics Protection in Zhejiang Province 2019008 and 2018007, Chinsese National Double First-rate Project about digital protection of cultural relics in Grotto Temple and equipment upgrading of the Chinese National Cultural Heritage Administration scientific research institutes.

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
