[Supplementary Material]

# Supplementary Material
# Learn, Imagine and Create: Text-to-Image Generation from Prior Knowledge

**Training details:** LeicaGAN was trained for 650 epochs using the ADAM optimizer with $lr = 0.0002$ on an NVIDIA Tesla V100 machine with 16GB memory. The training took around 2.5 days. The parameter information of each model in the paper are reported in Table 2. Following [21,37,43], for calculating the Inception score and R-precision, each model first generated 30,000 images conditioned on each testing set. Then we calculated the inception score and R-precision results.

Table 1: The number of parameters of each module in LeicaGAN.

| Module | Number of parameters |
|---|---|
| Image Encoder $E_V$ | 22.50M |
| Text-Image Encoder $E_T^I$ | 2.075M |
| Text-Mask Encoder $E_T^M$ | 2.075M |
| Generator $G$ | 7.618M |
| Discriminator $D$ | 97.596M |

**Generators:** Our CAG adopts a cascaded architecture, in which images are generated in multi-steps. Figure 1 shows the detailed generator structures for initial step ($i = 0$) and the following steps ($i = \{1, 2, ..m\}$). The **Attention Module** denotes the attention generation process detailed in Eq.(13) and (14) in the paper. Details about each layer are explained in Table 2.

Figure 1: Detailed structures of generators at different steps.

**Ablation studies and discussion:** More ablation study results are reported in the Table 3. According to our experiment results, it is notice-worthy that a higher Inception Score does not necessarily correspond to a higher R-precision score. We suspect that this may be because these two evaluation metrics evaluate the generated images according to different aspects, $i.e.$ the Inception Score evaluates

Table 2: Layer Information

| Module | Layer Information |
|---|---|
| FC | Fully Connected Layer |
| UpBlock | Upsampling$\times$2, Conv-(K3$\times$3, S1, P1), BN, GLU |
| Residual Block | Conv-(K3$\times$3, S1, P1), BN, GLU, Conv-(K3$\times$3, S1, P1), BN |
| Conv3x3 | Conv-(K3$\times$3, S1, P1), Tanh |
| BN | Batch Normalization |
| GLU | Gated Linear Unit |
| K, S, P | kernel, stride, padding |

diversity whereas the R-precision score evaluates semantic consistency. Take the Oxford-102 dataset as an example, this dataset contains many similar images with similar text descriptions. Therefore, a higher R-precision score for generated images means that these images semantically match the input text better, however, due to the high similarity among the text descriptions, it might lead to worse diversity and thus a lower Inception Score. We believe it would be valuable to explore more efficient modules to strengthen the impact of imagination and increase the diversity of the generated images, and a more objective evaluation method is also worthy to be explored in the future work.

Table 3: Other results of LeicaGAN trained with $E_T^I$ and $E_T^M$ of different loss weights on the CUB* and Oxford-102* datasets.

| Evaluation Metric | CUB* Dataset | |
|---|---|---|
| | Inception Score | R-precision [%] |
| (1) LeicaGAN, $\gamma_1 = 1, \gamma_2 = 0, \gamma_3 = 4, \gamma_4 = 1, \gamma_5 = 1, \gamma_6 = 0.5$ | 5.65$\pm$0.06 | 85.38 |
| (2) LeicaGAN, $\gamma_1 = 1, \gamma_2 = 1, \gamma_3 = 4, \gamma_4 = 1, \gamma_5 = 0, \gamma_6 = 0.5$ | 5.62$\pm$0.06 | 84.90 |
| (3) LeicaGAN, $\gamma_1 = 1, \gamma_2 = 1, \gamma_3 = 4, \gamma_4 = 1, \gamma_5 = 5, \gamma_6 = 5$ | 5.39$\pm$0.05 | 86.75 |
| (4) LeicaGAN, $\gamma_1 = 1, \gamma_2 = 1, \gamma_3 = 0, \gamma_4 = 1, \gamma_5 = 5, \gamma_6 = 5$ | 5.29$\pm$0.05 | 87.35 |

| Evaluation Metric | Oxford-102* Dataset | |
|---|---|---|
| | Inception Score | R-precision [%] |
| (1) LeicaGAN, $\gamma_1 = 1, \gamma_2 = 0, \gamma_3 = 4, \gamma_4 = 1, \gamma_5 = 1, \gamma_6 = 0.5$ | 3.65$\pm$0.03 | 78.92 |
| (2) LeicaGAN, $\gamma_1 = 1, \gamma_2 = 0, \gamma_3 = 4, \gamma_4 = 1, \gamma_5 = 0, \gamma_6 = 0.5$ | 3.72$\pm$0.03 | 77.60 |
| (3) LeicaGAN, $\gamma_1 = 1, \gamma_2 = 1, \gamma_3 = 4, \gamma_4 = 1, \gamma_5 = 0, \gamma_6 = 0.5$ | 3.60$\pm$0.03 | 79.60 |
| (4) LeicaGAN, $\gamma_1 = 1, \gamma_2 = 5, \gamma_3 = 0, \gamma_4 = 1, \gamma_5 = 0, \gamma_6 = 0.5$ | 3.64$\pm$0.02 | 86.37 |

the body is full of brown and white feathers with grey legs.

this bird is brown with white and has a very short beak.

the bird has a white breast and belly, and a small bill.

this bird is brown with black eyesand has a long beak.

this bird has a orange breast and belly and a black head.

this is a bird with dark gray wings and a lightcolored belly.

this bird is black with grey and has a very short beak.

the bird has a spotted back and curved tan bill is small.

a bright brown bird with a white beak and vibrant breast.

this is a fat bird with a small head and beak.

this bird has white wing bars and a grey chest.

the bird is small with a pointed bill, and the belly is yellow.

this particular bird has a belly that is white and black.

a bird with a cream belly, tan back, and yellow throat.

a medium sized bird with a white, brown, and gray body.

this bird has a blue head, with a orange breast.

this white bird has a yellow beak with a red dot.

this bird has wings that are blue and has a white belly

this bird is black and grey with a long tail and a white breast.

this bird has wings that are black and has a white crown

a small bird with a white belly, breast and eyering.

this small white bird has black wings and a small head.

this bird is white with brown and has a long, pointy beak.

the bird is small with a pointed bill and a red superciliary.

the bird has striped back and a black small bill.

this bird is brown with white and has a very short beak.

small yellow black and grey bird with small black beak.

this bird is black with white and has a long, pointy beak.

this bird is yellow, gray in color, with a dark beak.

a small red bird with black wings and a sharp black beak.

Figure 2: Examples of images generated by our LeicaGAN on the CUB testing set.

the petals of this flower are orange with a long stigma.

this flower has large white petals and long stamens.

this flower is purple and white, with petals are oval shaped.

this flower has petals that are pink with white stamen.

the flower petals are rounded and are yellow in color.

the petals of the flower are pink and the center is yellow.

a light pink flower with five petals and a pink pollen tube.

this flower has white petals with big pink lines.

this flower has petals that are purple with white stamen.

this flower is pink and white in color, with multi colored petals.

a flower with yellow petals and orange toward the stamen.

this flower has yellow and pink petals that have many layers.

this flower is yellow and pink, has multi colored petals.

the petals of this flower are blue with a short stigma.

this flower has red and white petals with white stamen.

this flower has big light yellow and orange.

this flower has petals with purple ,white ,yellow patches.

bright pink flower with big petals.

this flower has long yellow petals surround a dark center.

this flower has long pink petal which are thin with edges.

pretty white flowers with a yellow center.

the petals on this flower are pink with yellow stamen.

the petals of this flower are blue.

this flower is yellow in color, and has petals along the edges.

purple flower has large petals and white and purple pistils.

this white and pink flower has petals of different.

this flower has a ringed red and yellow petals.

this flower has very shiny red petals which are very rounded.

this flower is pink in color, with petals wrapped around.

this flower has light pink petal with a yellow anther filament.

Figure 3: Examples of images generated by our LeicaGAN on the Oxford-102 testing set.

Figure 4: Visualization of attention maps and images generated by LeicaGAN. The first row shows the generated images in three steps. The attention maps shown in yellow, red and green frames correspond to intermediate attention for producing $w_I^i$, $w_M^i$ and $\hat{s}_{IM}^i$ in Eq.(13) and (14).

Figure 5: **Human perceptual test:** To obtain higher-quality human perception study results, we recruited 60 undergraduate and graduate students. The first 5 trials of each test were considered as training. Then the remaining process consisted of *Real&Fake* and *Shape-Wise Test*, and *Pair-Wise Test*. Images generated by different methods were randomly shown to students. The interfaces of these tests are shown below.

|  | 0 | 1 | 2 | 3 | 4 | 5 |
|---|---|---|---|---|---|---|
| Real/Fake score | ○ | ○ | ○ | ○ | ○ | ○ |
| shape-wise score | ○ | ○ | ○ | ○ | ○ | ○ |

* white bird with black stripes across the top of the head and the tail.

|  | 0 | 1 | 2 | 3 | 4 | 5 |
|---|---|---|---|---|---|---|
| Pair-wise score | ○ | ○ | ○ | ○ | ○ | ○ |