[Reviews · NeurIPS 2019]

Reviewer 1



Quality The paper is thorough in describing the method and supporting the proposed method with experiments Clarity The paper is well written and easy to follow. Originality & Significance Although the method is not very novel in light of Paper 1253: RecreateGAN (see more below), the experimental exploration of different settings of the method is thoroughly done and interesting. The idea of matching local image features to word-level embeddings and matching global image features to sentence level embeddings is intuitive and makes sense. This paper shares significant parts of the method with Paper 1253: RecreateGAN, in particular, the textual-visual embedding loss in (6) of this paper matches the pairwise loss defined in eq (5) of the other paper. However, this paper uses this component as part of a different method, namely for textual-visual embedding vs an image similarity embedding. Additionally, the cascade of attentional generators in this papers is very similar between both papers. Although both papers can be seen as methods to translate something to an image (in this case text, in the other paper, an image) using similar embedding methods to condition a cascade of attentional generators, the details of the methods and tasks still have quite a bit of differences between each other, and it would not be possible to explain both systems in one 8-page paper. I find the text-to-image paper more compelling, as the task itself makes more sense to me, and the results are state of the art. However, as there are similarities, I am repeating some comments from my review of the other paper here: 1. The rationale for the attention-weighted similarity between two vectors is unclear to me. Since all the loss in (6) is symmetric in w and l (and also symmetric in s and g), why not use a symmetric similarity measure? I would also appreciate an exploration of the effect of using a cosine similarity here. 2. I don't understand why the probability of w being matched by l is calculated over the minibatch in the formulation of (3). Why is it desirable to have p(l | w) depend on the other members of the batch?

Reviewer 2



I agree the motivation of "learn, imagine and create". The whole model is intuitively simple and easy to implement. But I have a few concerns. First, since the Text-image encoder directly embeds the sentence, how can the model be able to control each word as presented in the experiments? Second, as far as I know, there are so a lot of works on text-image generation. Finally, the useage of the concept "prior knowledge" is very inappropriate. The prior knowledge ofter refers to the symbolic knowledge which can be interpretable and explained the reason. But this paper only named the text-image encoded feature as the prior knowledge, which is overclaimed. And the claims about "mimicing human imagination" is also not good as it is just a fusion strategy. If we think this paper as a practical model , it is fine. But the authors used too much overclaimed phrases to try to highlight some contributions which are not true.This paper only compares with AttGAN and lacks of enough comparison with recent works. Finally, the ablation studies to validate each key component is missing and not enough to support the contribution of this simple model.

Reviewer 3



his paper presents a novel text to image (T2I) generation method called LeciaGAN which is inspired by the process humans follow to achieve the same goal. These processes are modelled by decomposing the required tasks into three phases: the knowledge learning phase, the imagination phase, and the creation phase. Semantic consistency and visual realism is achieved by using adversarial learning and cascaded attentive generator. The paper provides a good overview of the literature in the related works, and provides a good motivation for the proposed algorithm. Despite the many concepts required to clearly describe the model in few pages, the well-thought structure of the paper makes it easy to follow, while sufficient description is given to highlight the necessary elements. The metric used to (Inception Score and R-precision for objective measures, and subjective visual measures and human perceptual testing) is sufficient to establish the performance of the proposed algorithms as well as to compare it with already existing methods. I am quite satisfied and impressed by the result obtained and the method used to get there. I believe this paper has a significant contribution to the T2I field. Some minor comments are as follows: * The descriptions in Text-Image Encoder paragraph in section 3.1, line number 123, are slightly cryptic. I think rewriting it to clearly present the pairwise similarity part would greatly help the readability of this section. * Equation 7, just below line 141, the second term in the right side of the equation should be log(1-D_modal(s_n))

[Author Response · NeurIPS 2019]

We authors are grateful to the reviewers for their valuable comments. We will improve the final version by taken all the review comments and release the source code package to ensure the reproducibility. Below, we number and address comments of each reviewer in order.

Figure 1: Illustration of matched & mismatched $(T_n, I_n)$ pairs in a minibatch.

Table 1: Results of LeicaGAN trained with $E_T^I$ and $E_T^M$ of different loss weights.

| Evaluation Metric | Inception Score | | R-precision [%] | |
|---|---|---|---|---|
| | CUB* | Oxford-102* | CUB* | Oxford-102* |
| (1) LeicaGAN, $E_T^I(\alpha_1{=}0)$, $E_T^M(\alpha_2{=}0)$ | 5.51±0.03 | 3.64±0.01 | 82.89 | 84.02 |
| (2) LeicaGAN, $E_T^I(\alpha_1{=}1)$, $E_T^M(\alpha_2{=}0)$ | 5.58±0.05 | 3.71±0.02 | 85.20 | 85.55 |
| (3) LeicaGAN, $E_T^I(\alpha_1{=}5)$, $E_T^M(\alpha_2{=}0)$ | 5.60±0.05 | 3.68±0.01 | 82.95 | 84.03 |
| (4) LeicaGAN, $E_T^I(\alpha_1{=}1)$, $E_T^M(\alpha_2{=}2)$ | 5.63±0.04 | 3.79±0.01 | 84.57 | 84.98 |
| (5) LeicaGAN, $E_T^I(\alpha_1{=}1)$, $E_T^M(\alpha_2{=}4)$ | **5.69±0.05** | **3.80±0.01** | **85.28** | **85.81** |
| (6) LeicaGAN, $E_T^I(\alpha_1{=}1)$, $E_T^M(\alpha_2{=}6)$ | 5.61±0.06 | 3.64±0.02 | 81.65 | 84.01 |

**R1.1 -** Why is the probability of $w$ matched by $l$ calculated over the minibatch in Eq (3)?

$\Rightarrow$ Assuming a minibatch of 4 pairs of $(T_n, I_n)$, where $n \in \{1, 2, 3, 4\}$ as shown in Fig.1, we treat the green and red pairs as matched and mismatched pairs respectively. Therefore, for each text, we obtain 4 similarity scores indicating the closeness between the given text with the matched & mismatched images (vice versa). Afterwards, according to Eq.(4), $p(w|l)$ and $p(l|w)$ are estimated separately. Specifically, a softmax is applied among the matched and mismatched text-image (image-text) pairs in a batch along the row (column) as shown in Fig.1. They serve as different constraints for training the model at the same time.

**R1.2 -** Why not use a symmetric similarity measure? An exploration of the effect of using a cosine similarity.

$\Rightarrow$ The local-level similarity score $s_{w|l}$ or $s_{l|w}$ of the (T, I) pair is first calculated as: $log(\sum_{i=1}^{L} exp(\gamma_1 cos(\widehat{l_i}, w_i)))^{1/\gamma_1}$ [37], which is a symmetric similarity result and we elaborate this equation here for a more clear explanation of the calculation. Careful revision of this part and further exploration about different similarity measures will be included in the final version.

**R1.3 -** The imagination phase is just sampling a noise vector.

$\Rightarrow$ The embedded textual features obtained in TVE convey different kinds of visually-relevant information. Therefore, in the imagination phase, an aggregation of these features and noise is applied to obtain an initial visual impression serving as the input for the subsequent creation phase. The noise is added to guarantee the diversity of the image generation. In the final version, we will explore more efficient and diverse modules to strengthen the impact of the imagination phase as we discussed in the section of Limitation.

**R1.4 -** What is the effect of having the adversarial term in the encoder loss? An ablation study of this adversarial term.

$\Rightarrow$ The loss weights $\alpha_1$ and $\alpha_2$ in training text-visual co-embedding models $E_T^I$ and $E_T^M$, respectively, were carefully studied in Table 1. The detailed results will be reported in the final version. The comparison between (1) and (2) shows that the use of the adversarial loss obtains a performance gain. An intuitive illustration of the effect of the adversarial loss reducing the multi-modal domain gap has been shown in Fig.4 of [35].

**R2.1 -** How can the model be able to control each word as presented in the experiments?

$\Rightarrow$ The input sentence was first embedded into both (1) word-level feature matrices $w$ to represent the semantic meaning of the words and (2) a sentence-level feature vector $s$ to convey the semantic of the whole sentence. Afterwards, through the co-embedding of images and text, the correlations between them can be built. Therefore, during the decoding stage, with the guidance of the attention module, the generators focused on different words, resulting in different visual outputs accordingly.

**R2.2 -** This paper only compares with AttnGAN and ablation studies of key components are missing.

$\Rightarrow$ Before the NeurIPS submission deadline, StackGAN and AttnGAN were recognized as state-of-the-art models in T2I. Recently, Obj-GANs, SD-GAN, MirrorGAN, and DM-GAN are published in CVPR2019. However, it is not straightforward to compare LeicaGAN with all of them because they were tested on different datasets using different evaluation metrics and only two of them (MirrorGAN and Obj-GAN) released their source code packages in the last two months, after the NeurIPS submission deadline. To make a fair comparison, we will re-train and test these models under the same condition in the final version. Based on the results in Table 2 in the paper, we verified the effectiveness of the following key components: (1) the global-local attentive generator, (2) $E_T^M$, and (3) the collaborative attention module and its weights. Here, we report the ablation study for the weight choosing of the TVE models in Table 1. More details will be provided in the appendix and code.

**R2.3 -** The paper names the text-image encoded feature as the prior knowledge, which is overclaimed.

$\Rightarrow$ We name the text-image encoded features as the prior knowledge because the co-embedding mappings, which correlate textual and visual features with each other, are learned in the PKL phase. These cross-modal embeddings serve as the prior knowledge to guide the image generation of the subsequent phases. We can expect that this phase can be further improved by incorporating different types of prior. We agree the name of "prior knowledge" can be inappropriate and will change it to "prior" in the final version.

**R3.1 -** Improving the limitations the authors already discussed in the paper.

$\Rightarrow$ In the future, we plan to introduce more prior to the first PKL phase, such as fine-grained attributes, and more efficient and diverse modules will be explored to strengthen the impact of imagination. Additionally, more advanced network structures and training strategies will be employed in order to further enhance the generation ability of the proposed T2I model.

[Meta-Review · NeurIPS 2019]

This paper proposes a new method for text to image generation. Pros • A new method is proposed. • State of the art results are obtained for the task. • The paper is generally clearly written. • The proposed method is reasonable and sound. Cons • There are several issues pointed out by the reviewers. Some of them have been addressed in the author response. • Comparison with some existing methods is missing. The authors have promised to add more. After the author response, the reviewers made a discussion and agreed that the paper can be a strong one if the issues are addressed in the next version.